# Talking in primary care (TIP): protocol for a cluster-randomised controlled trial in UK primary care to assess clinical and cost-effectiveness of communication skills e-learning for practitioners on patients' musculoskeletal pain and enablement

Felicity L Bishop [1], Nadia Cross [2], Rachel Dewar-Haggart [1,2]
Emma Teasdale [1,2], Amy Herbert [3], Michelle E Robinson [4]
Matthew J Ridd [5], Christian Mallen [4], Lorna Clarson [4]
Jennifer Bostock [2], Taeko Becque [2], Beth Stuart [6], Kirsty Garfield [7]
Leanne Morrison [1,2], Sebastien Pollet [1,2], Jane Vennik [2]
Helen Atherton [2,8], Jeremy Howick [9,10], Geraldine M Leydon [2]
Jacqui Nuttall [11], Nazrul Islam [2], Paul H Lee [11], Paul Little [2]
Hazel A Everitt [2]

For numbered affiliations see end of article.

**Correspondence to**
Professor Felicity L Bishop;
F.L.Bishop@southampton.ac.uk

## ABSTRACT

**Introduction** Effective communication can help optimise healthcare interactions and patient outcomes. However, few interventions have been tested clinically, subjected to cost-effectiveness analysis or are sufficiently brief and well-described for implementation in primary care. This paper presents the protocol for determining the effectiveness and cost-effectiveness of a rigorously developed brief eLearning tool, EMPathicO, among patients with and without musculoskeletal pain.

**Methods and analysis** A cluster randomised controlled trial in general practitioner (GP) surgeries in England and Wales serving patients from diverse geographic, socioeconomic and ethnic backgrounds. GP surgeries are randomised (1:1) to receive EMPathicO e-learning immediately, or at trial end. Eligible practitioners (eg, GPs, physiotherapists and nurse practitioners) are involved in managing primary care patients with musculoskeletal pain. Patient recruitment is managed by practice staff and researchers. Target recruitment is 840 adults with and 840 without musculoskeletal pain consulting face-to-face, by telephone or video. Patients complete web-based questionnaires at preconsultation baseline, 1 week and 1, 3 and 6 months later. There are two patient-reported primary outcomes: pain intensity and patient enablement. Cost-effectiveness is considered from the National Health Service and societal perspectives. Secondary and process measures include practitioner patterns of use of EMPathicO, practitioner-reported self-efficacy and intentions, patient-reported symptom severity, quality of life, satisfaction, perceptions of practitioner empathy and optimism, treatment expectancies, anxiety, depression and continuity of care. Purposive subsamples of patients, practitioners and practice staff take part in up to two qualitative, semistructured interviews.

## STRENGTHS AND LIMITATIONS OF THIS STUDY

⇒ Assessment of a brief online learning package that is evidence-based and theory-based and was rigorously developed with primary care clinicians.

⇒ Practitioners (eg, general practitioners, physios and nurses) consult as usual without needing to identify or obtain consent from patients within the consultation, as patient recruitment is done by administrative staff.

⇒ Focused on patients with musculoskeletal pain but including other patients as 'all-comers' enables an efficient test of relevance to all primary care consultations.

⇒ Feasibility work showed it is not practicable to record consultations in this trial, so there is no direct assessment of changes in practitioner communication behaviours after engaging with the e-learning package.

⇒ 'All-comers' is a large and varied group of patients that enhances generalisability but is not suitably powered to plan subgroup analyses.

**Ethics approval and dissemination** Approved by the South Central Hampshire B Research Ethics Committee on 1 July 2022 and the Health Research Authority and Health and Care Research Wales on 6 July 2022 (REC reference 22/SC/0145; IRAS project ID 312208). Results will be disseminated via peer-reviewed academic publications, conference presentations and patient and practitioner outlets. If successful, EMPathicO could

quickly be made available at a low cost to primary care practices across the country.

**Trial registration number** ISRCTN18010240.

## INTRODUCTION

Approximately 1.7 billion people worldwide have musculoskeletal conditions, which are typically painful, limit people's daily lives and impair quality of life.[1] Musculoskeletal conditions, including back, hip, knee and neck pain, are commonly managed in primary care,[2–4] where patient-centred care, including excellent practitioner–patient communication, is an international priority.[5–7] In the UK, people with musculoskeletal conditions may be seen in primary care by general practitioners (GPs), practice nurses, physiotherapists and other allied healthcare professionals.

Regardless of which treatment, therapy or other intervention a patient receives, effective practitioner–patient communication can reduce symptoms and enhance quality of life, adherence to and satisfaction with care, producing benefits comparable to many pharmaceutical interventions.[8–10] Suboptimal communication can lead to missed opportunities for benefit, worse quality of life and symptom management, unwanted prescriptions and non-adherence,[11 12] unnecessary economic costs,[12] deviations from guideline-recommended treatment[13] and increased complaints and litigation.[14 15] Despite communication skills being taught in medical and allied health professional training, patients still report dissatisfaction with practitioner–patient communication.[16 17] The extent to which patients rate their practitioners as being empathic varies widely[18] and medical students appear to exhibit broadly stable or declining levels of empathy during their degrees.[19 20] The need to enhance and expand communication skills is particularly pertinent since the COVID pandemic forced the rapid introduction of remote consultations, bringing new opportunities and challenges for patients and staff not specifically trained to consult in this way.[21]

We focus on the communication of clinical empathy and positive messages within primary care consultations. Clinical empathy and positive messages are not routinely reliably optimised in clinical care but can have statistically and likely clinically significant effects on pain, patient satisfaction and other outcomes with no evidence of adverse effects.[22] Our intervention planning determined that enhancing practitioners' communication of clinical empathy and realistic optimism was feasible, measurable and likely to have a significant impact.[23 24] Even brief interventions can improve communication skills, including interventions concentrating on empathy skills such as active listening and expressing warmth at appropriate times,[25–27] which take no additional time in the consultation.[27 28] However, few interventions have been tested clinically for effects on patients' health,[29] subjected to formal cost-effectiveness evaluations[30] or are sufficiently brief and well-described to facilitate implementation in the current primary care climate. Our work aims to address these limitations. We are evaluating the effects on patients' health of brief, evidence-based, online training to enhance practitioners' communication of clinical empathy and realistic optimism within everyday clinical consultations ('EMPathicO').

## AIMS AND OBJECTIVES

The primary objective is to determine EMPathicO's effects on (1) patient-reported pain and (2) patient enablement via repeated measures over 6 months following the index consultation in patients presenting with musculoskeletal pain, compared with usual care control.

This clinical focus on musculoskeletal pain was chosen to align with the EMPathicO training, which includes modules on clinical empathy, realistic optimism and how to communicate these better in the context of consultations for osteoarthritis. Including a condition-specific module permitted a clear demonstration of communication skills in a particular context, which made the training better targeted and potentially more effective.[31] A painful musculoskeletal condition was chosen because much (but not all) of the evidence that underpins the importance of clinical empathy and realistic optimism for patient outcomes is derived from studies of pain and painful conditions; osteoarthritis was chosen because it is a prevalent painful musculoskeletal condition in primary care.

Secondary objectives are as follows.

► To estimate EMPathicO's cost-effectiveness and effects on patient-reported quality of life and other secondary outcomes, over 6 months from index consultation, in patients with musculoskeletal pain.
► To test hypothesised mechanisms of action.
► To explore EMPathicO's potential for implementation, by:
  – Determining EMPathicO's effects on patient enablement, patient-reported quality of life and other secondary outcomes over 6 months from index consultation, in patients ineligible for the musculoskeletal pain group (ie, presenting with other symptoms and/or very low levels of musculoskeletal pain, hereafter referred to as 'all-comers'). This group was included because clinical empathy and realistic optimism may be beneficial for many different symptoms seen in primary care, and when practitioners adopt new communication behaviours within consultations for one type of condition, these skills may 'spill over' and also be implemented in consultations for other conditions. We wanted to evaluate any such additional benefits.
  – Identifying opportunities, barriers and solutions for widespread implementation and impact, using the RE-AIM framework to explore EMPathicO's Reach, Effectiveness, Adoption, Implementation and Maintenance.[32 33]

## METHODS AND ANALYSIS

This protocol is reported in accordance with the Standard Protocol Items: Recommendations for Interventional Trials checklist (online supplemental material 1).[34] The first site was randomised on 31 October 2022, and data collection is due to finish on 31 July 2024.

### Patient and public involvement

To ensure our work engages and is relevant to patients, we have worked with patients and members of the public throughout the development of EMPathicO and this protocol. We continue working closely with our Patient Advisory Group, led by our patient and public involvement and engagement lead (PPIE), JB, who sits on our trial management group. Our Patient Advisory Group comprises six patient and public contributors of varying ages, ethnic backgrounds (three from black and minority ethnic backgrounds and three from white backgrounds), gender (three females and three males) and geographical locations within England. One member is neurodivergent, and all have lived experience of musculoskeletal pain as patients or carers. Our panel meets virtually for 1 hour bimonthly and contributes to specific activities, including refining patient-facing documents and procedures, training qualitative interviewers and interpreting data.

### Design

A cluster-randomised controlled parallel group superiority trial in primary care, with embedded qualitative and mixed-methods process and implementation analyses.

Cluster randomisation was chosen because randomising individual practitioners risks cross-contamination within practices where practitioners share knowledge and patients; randomising individual patients risks contamination because practitioners cannot switch on/off communication skills in different consultations.

General practices constitute the clusters; practices are recruited and then randomised 1:1 EMPathicO:control. Randomisation is stratified (see below). All eligible practitioners within clusters are encouraged to undertake EMPathicO training (intervention) or consult patients as usual (control). The control was chosen to enable a pragmatic assessment of the benefits and costs of adding EMPathicO training to usual care.

Patient recruitment commences at least 2 weeks after the general practice is randomised (enabling time for intervention sites to complete the intervention training while maintaining consistent set-up timelines across both arms). All adults (18+) verbally consulting a participating practitioner are invited to participate in the trial (see exclusions below).

Two groups of patients are recruited. The musculoskeletal group comprises patients consulting participating practitioners about musculoskeletal pain. The 'all-comer' group comprises patients consulting about symptoms other than musculoskeletal pain (or reporting very low levels of musculoskeletal pain). At preconsultation baseline and repeatedly up to 6 months later, patients complete questionnaires assessing pain, enablement and secondary outcomes.

### Setting

General practices in England and Wales are recruited and supported by three recruitment hubs: Southampton, Keele and Bristol.

### Target population

#### GP practice eligibility criteria

*Eligible*: National Health Service (NHS) general practices in England and Wales, where a general practice is 'an organisation which offers primary care medical services by a qualified general practitioner who can prescribe medicine and where patients can be registered and held on a list'.[35]

*Excluded*: practices involved in intervention development and feasibility work (18 from Wessex and 5 from the West Midlands), practices where clinical members of the trial management group or trial steering committee (TSC) see patients.

#### Practitioner eligibility criteria

*Eligible*: practitioners from any discipline who are working within participating GP surgeries and seeing patients with musculoskeletal pain (eg, GPs, practice nurses, physiotherapists, pharmacists and physician associates).

*Excluded*: practitioners unwilling to undertake the intervention or trial procedures.

#### Patients with musculoskeletal pain eligibility criteria

For the musculoskeletal pain group, eligible patients are adults (18+); verbally consulting a participating practitioner about new, recurrent or ongoing musculoskeletal pain (eg, back, hip, upper or lower extremity, neck pain—consistent with the International Classification of Diseases-11th edition's diseases of the musculoskeletal system[36]); reporting average pain in the last week as four or more on a numerical rating scale at baseline (0=no pain; 10=pain as bad as you can imagine); consulting face-to-face, telephone or videoconference and able to give informed consent. The first consultation is the 'index' consultation, an initial triage interaction does not constitute an 'index' consultation. People without English as a first language are eligible; interpreters are available to support access to trial paperwork and patient-reported measures, and their use is recorded; informal interpreters (eg, family) may also provide support.

*Excluded*: patients consulting solely in written forms (eg, e-consult or email), pain caused by malignancy, being unable to consent or to complete questionnaires (eg, severe mental illness or distress and terminal illness), already enrolled in the trial (ie, from a previous consultation) and aged<18.

#### All-comer patients' eligibility criteria

For the all-comers group, eligible patients are adults (18+); verbally consulting a participating practitioner

about something other than musculoskeletal pain or consulting for musculoskeletal pain and rating average pain in the last week as less than four at baseline and able to give informed consent.

*Excluded*: as for patients with musculoskeletal pain.

### Interventions

#### EMPathicO e-learning package

EMPathicO is an evidence-based, theoretically grounded digital e-learning package for practitioners routinely seeing patients frontline in primary medical care, including GPs, nurse practitioners and first-contact physiotherapists.[24] EMPathicO helps practitioners enhance their communication of clinical empathy and realistic optimism, is consistent with major consultation models including Ideas, Concerns and Expectations (ICE)[37] and incorporates behaviour change techniques. Using the Behaviour Change Wheel, EMPathicO was designed to target users' motivation (reflective and autonomic), capability (physical and psychological) and opportunity (environmental) through the intervention functions of persuasion, incentivisation, enablement, education, training, modelling and environment restructuring. Multiple behaviour change techniques were used to achieve these functions, including demonstration, information provision, goal-setting, action planning and instruction. For a complete behavioural analysis of EMPathicO, see supplementary material in our intervention development paper.[24]

The brief interactive e-learning modules are completed by practitioners and can be completed separately or together in less than 75 min. They cover clinical empathy, realistic optimism, tailoring empathy and optimism for patients with osteoarthritis (a common cause of musculoskeletal pain), evaluating one's own consultations and goal-setting. The structure and contents of the modules

are summarised in figure 1. EMPathicO was developed using LifeGuide open-source software for creating online interventions for healthcare, health promotion and training.[38]

The systematic process of developing EMPathicO using the person-based approach[39] involved multiple literature reviews, behavioural analysis and extensive iterative qualitative research.[40–46] This work all contributed to the underpinning logic model (figure 2).[24]

#### Control: usual care

Practitioners in practices randomised to usual care control do not receive training and are asked to consult as usual. They are offered access to EMPathicO after all patient recruitment and follow-up are completed.

#### Concomitant interventions

All practitioners are discouraged from undertaking additional communication skills training during the study and must self-report any that does occur.

### Recruitment

#### GP practice recruitment

Practices are recruited with local Clinical Research Network support, seeking practices of different sizes (small and large) and locations (urban and rural) and those serving populations in areas of higher deprivation and greater ethnic diversity.

#### Practitioner recruitment

Practitioners within participating practices are recruited by that practice's lead for this study (the local principal investigator) with support from the trial team and materials including an infographic and 1-min video explaining the study.

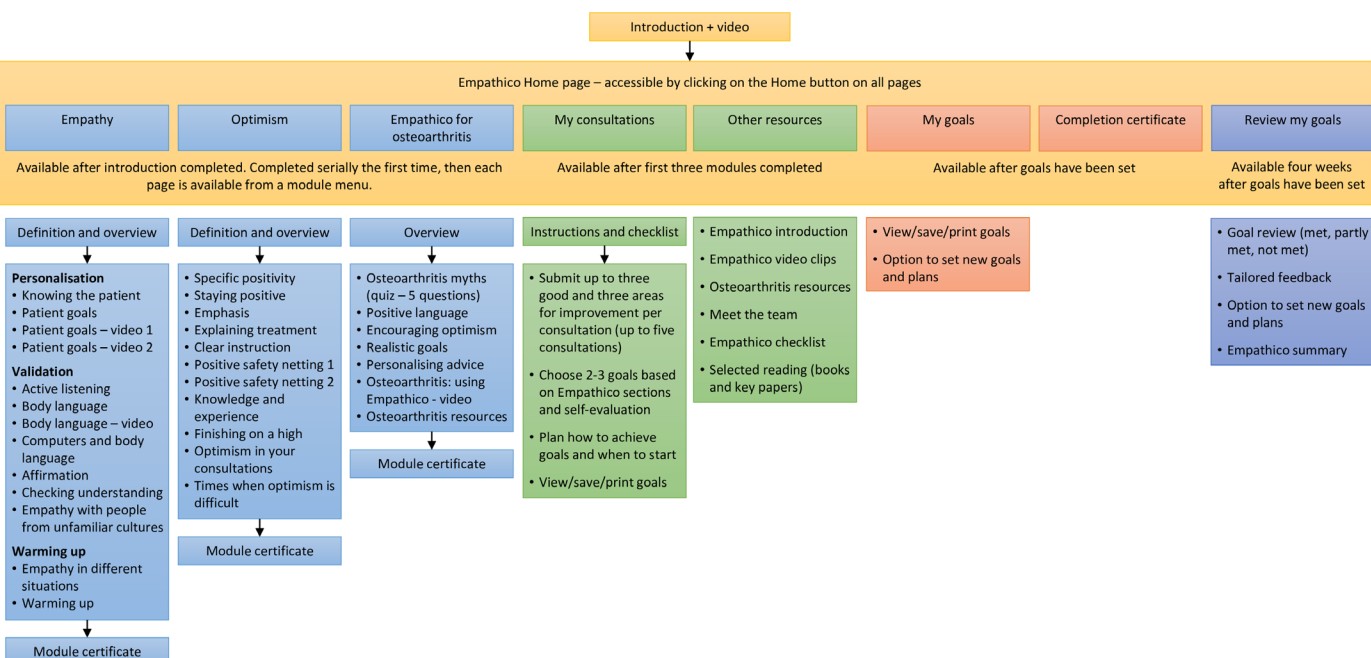

**Figure 1** Schematic summary of EMPathico structure and contents.

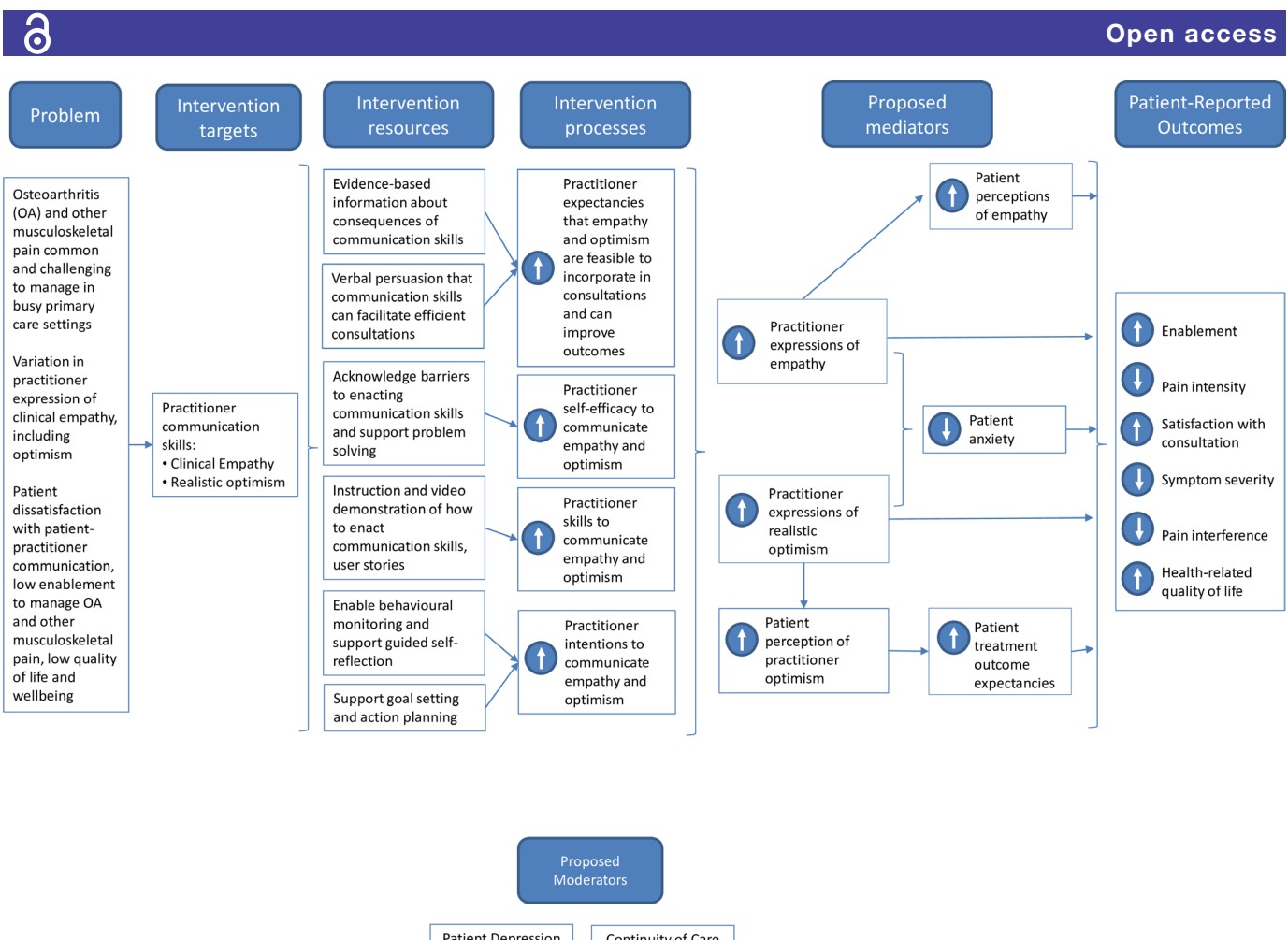

**Figure 2** Logic model showing how EMPathicO is hypothesised to affect patient outcomes.

## Patient recruitment

Practices invite into the study consecutive patients who are consulting participating practitioners within the recruitment period, after screening out any patients who do not have the capacity for consent or where there are medical grounds for excluding the patient (eg, very unwell generally and severe mental distress). Patient recruitment methods are tailored to suit individual practices' appointment booking systems. For patients with prebooked or same-day appointments, practices text, email or post a brief invitation and link to the patient-facing study website up to 1 week before their consultation. Practices screen potential invitees for initial eligibility before sending invitations. Practices may display a poster in practice and/or on their website. Reception staff may introduce the study to patients attending in person. Patients email or phone the patient-facing research team with questions.

Practices follow their usual procedures for contacting non-English speakers to invite them to take part, for example, contacting a designated friend, relative or support worker, arranging an interpreter or adding a sentence in the patient's own language on the initial study invitation.

The number of patient invitation emails or texts sent by each site is collected and recorded centrally. Qualtrics records instances of patients accessing the study website but declining consent and/or not meeting inclusion criteria.

The patient-facing study website is hosted on Qualtrics and shows the full study invitation and patient information sheet (PIS) (in languages requested by practices). After reading the PIS, patients complete a brief screening questionnaire, online consent and baseline measures. Online supplemental material 2 contains PIS and consent forms.

## Sample size

### Patients with musculoskeletal pain sample size

The minimum clinically important difference in the pain primary outcome is approximately one point,[47] SD 3.3, consistent with a standardised effect size of 0.3. For 90% power, an alpha of 0.025 to allow for two primary outcomes, and a correlation between the four repeated measures of 0.7, a sample size of 214 per group is required. We assume a conservative intraclass correlation coefficient (ICC) of 0.03, at the upper 75% of what has been observed in previous primary care trials.[48] Assuming 20 patients per practice gives a design effect of 1.57. Allowing for a 20% loss to follow-up gives a total sample size

of $(214*2*1.57)/0.8=840$ participants to be recruited from 42 practices.

### 'All-comer' patients' sample size

Recruiting 840 all-comers will give 90% power (based on alpha and ICC as per the musculoskeletal group above) to detect a standardised effect size of 0.3 in the enablement primary outcome, equivalent to a difference of 0.36 points (assuming SD=1.2[49]).

### Updated sample size calculation

Participants are being recruited from 53 practices rather than 42 practices as originally planned, which reduces the average cluster size. Assuming 14 patients per practice gives a design effect of 1.39. Under the same assumptions as above, the total sample size is $(214*2*1.39)/0.8=744$ participants.

## Outcomes

### Questionnaires, data collection and participant retention

Online supplemental material 3 summarises outcome and process variables, measurement timings and questionnaire measures. We considered core outcome sets, questionnaire properties (eg, validity, reliability and length) and acceptability to participants when choosing specific measures.

Patient-reported measures are completed on web-based questionnaires hosted on Qualtrics (Qualtrics, Provo, UT, USA); to support inclusive access, patients may request an interpreter and/or paper versions. £10 vouchers are sent at 1-month and 6-month follow-ups to incentivise completion.

Practitioner-reported measures are completed on LifeGuide[38] (measures completed by the intervention group only) and Qualtrics (measures completed by all practitioners).

For practitioners and patients, automated follow-up emails are sent to non-responders at all timepoints. Researchers personally contact persistent non-responders who haven't withdrawn and offer to resend questionnaires or complete primary outcomes by telephone.

## Primary outcomes

For the musculoskeletal pain group, the two primary outcomes are pain intensity and patient enablement, each analysed over 6 months using a repeated measures approach. Pain intensity is the severity of the pain sensation and is included in the core outcome sets for chronic pain,[50 51] osteoarthritis[52] and low back pain.[53 54] Patient enablement refers to patients' feelings, after a consultation, of confidence and empowerment to cope with their symptoms, to stay healthy and to help themselves. Our PPIE work highlighted enablement as at least as important as pain. Two primary outcomes help capture more holistic effects on patients' health. The outcomes will be reported separately, and our PPIE and embedded qualitative work will help explore, interpret and explain how they relate to each other.

For the all-comers group, patient enablement is the single primary outcome. Pain intensity is measured as a secondary outcome if pain is present.

### Pain intensity

Pain intensity is measured as average pain in the last week using the four-item pain intensity subscale from the Brief Pain Inventory (BPI).[55]

### Patient enablement

The six-item Patient Enablement Index (PEI) captures patients' feelings, after a consultation, of confidence and empowerment to cope with their symptoms, to stay healthy and to help themselves.[56] To increase sensitivity, versions with more response options than the original four (much better/never/same or less/not applicable) have been reported.[57–59] Following our feasibility study, we use a modified seven-point agree-disagree Likert response scale with a not applicable option.

## Secondary outcomes

### Symptom severity and global impression of change

Overall perceptions of symptom severity and change are important for musculoskeletal patients given the high prevalence of multimorbid conditions and for all-comers because they apply to any condition and provide a symptom-focused preconsultation baseline. Two single-item seven-point[60] measures of patient global impression of symptom severity and patient global impression of change are collected.[61]

### Patient satisfaction

The version of the 21-item Medical Interview Satisfaction Scale,[62] adapted and revalidated for UK primary care,[63] is used to measure patient satisfaction with the consultation.

### Pain interference

Pain interference is measured with the seven-item pain interference scale from the BPI.[55]

### Health-related quality of life

Health status is measured using the five-item EuroQoL five-dimension five-level (EQ-5D-5L) and the EQ-visual analogue scale.[64]

## Health economics outcomes

Cost-effectiveness will be assessed from NHS and societal perspectives, including personal expenses and productivity, over 6 months. Utility values will be estimated from EQ-5D-5L scores using National Institute for Health and Care Excellence (NICE)-recommended approach at the time of analysis. Quality-adjusted life-years will be estimated by combining utility values, with the length of time in each health state, using the area under the curve approach.[64–66] The five-item ICEpop CAPability measure for Adults (ICECAP-A), which was designed to capture broader aspects of quality-of-life and has been found to complement the EQ-5D in economic evaluations, is also collected.[67 68]

Practitioner time spent on EMPathicO training is captured by LifeGuide. Resource-use data are collected using ModRUM[69] (patient self-reported healthcare utilisation) and bespoke questions (costs outside the healthcare sector, eg, personal expenses). The Work Productivity and Activity Impairment Questionnaire: General Health is used to collect information on productivity, including time off work.[70] NHS resources include primary, community and secondary care and prescribed medications; they will be valued using the national unit costs.[71–73] Personal expenses will be presented as reported. Sick leave from employment will be valued using the annual survey of hours and earnings.[74]

### Process variables and covariates

Potential mediators and moderators of intervention effects on pain, specified in the logic model, are included as process variables. Practitioner-reported self-efficacy, outcome expectancy and intentions for conveying empathy and optimism in consultations are assessed using bespoke items developed in our feasibility work based on standard item stems, relevant guidelines and theory.[75–78] They demonstrated acceptable internal consistency (Cronbach's alphas ranged from 0.69 to 0.98) and were fully completed by practitioners (n=11).

Intervention usage data captured on LifeGuide includes, for each practitioner and participant, time spent on (different sections of) the intervention and patterns of access.

Patient perceptions of practitioner clinical empathy are assessed using the 10-item consultation and relational empathy (CARE) scale[79] used extensively in UK primary care settings to assess patient perceptions of clinical empathy. Patient perceptions of practitioner response expectancies are assessed using a bespoke single item tested in our feasibility study. Patient treatment outcome expectancies are measured using the 15-item six-subscale, Treatment Expectation Questionnaire.[80] Patient anxiety and depression are assessed using the seven-item subscales from the Hospital Anxiety and Depression Scale (HADS).[81 82] Continuity of care is assessed using the nine-item Patient–Doctor Depth of Relationship Scale,[83] modified for non-doctor practitioners.

Practitioner characteristics collected are age, gender, ethnicity, years qualified and profession. Practice-level data collected from the practice and supplemented with data from national general practice profiles (National General Practice Profiles - Data – OHID, phe.org.uk) are list size, deprivation score and staffing.

Patient characteristics collected are age, gender, ethnicity, postcode (for calculating the index of multiple deprivation (IMD)), reason(s) for consulting (coded using the International Classification of Primary Care, second edition), comorbidities and index consultation modality.

### Qualitative interviews

A subsample of patients (up to n=45 with musculoskeletal pain and n=45 all-comers) and practitioners (up to n=40) take part in qualitative, semistructured telephone interviews. Participants are purposively sampled to capture diversity in index-consultation mode (telephone, video or face-to-face), ethnicity, age, gender, and baseline pain severity. Participants are interviewed twice each to explore short-term and long-term implementation of EMPathicO skills (practitioners) and experiences of the index and subsequent consultations (patients). Practitioners are interviewed after (1) patient recruitment and (2) follow-ups are completed at their practice. Patients are interviewed within approximately 7–14 days of their index consultation and again approximately 6 months later. Topic guides comprising open-ended questions and prompts are used flexibly and modified iteratively as necessary to explore emerging avenues of inquiry within the scope of the trial. Field notes are taken, interviews are transcribed verbatim, identifying details are replaced (eg, using pseudonyms), and transcripts are checked and imported to NVivo (Lumivero, Denver, CO, USA) for analysis.

### Timelines

Practitioner and patient timelines for enrolment, questionnaires and interviews are shown in tables 1 and 2.

### Assignment of interventions

#### Sequence generation, allocation concealment and implementation

A computer-generated allocation sequence is used with random block sizes of four and six. Blocks are stratified by practice-level high/low deprivation (IMD 1–5/IMD 6–10) and large/small practice size (list size >7900/<7900; 7900=median practice list size in England). The allocation sequence is implemented using the randomisation function in LifeGuide and is not visible to users. The trial manager (or their delegate) inputs each eligible practice to the randomisation function on LifeGuide which then displays the allocation. Practitioners and patients can withdraw from the study without giving a reason, but they cannot request modification to their allocated intervention.

#### Blinding

Patients and the trial statistician are masked to intervention allocation. Patients are not told in the PIS that, as part of this study, their general practice has been randomly allocated to intervention or control. This was approved by the ethics committee and is appropriate in this cluster-randomised trial where the communication skills training intervention is very low-risk and within the broad scope of usual practice. After all data collection is complete, patients will be debriefed in writing (email/mail) and told that 'at the start of the TIP study, some of the GP practices taking part had communication skills training (intervention practices), and some GP practices did not have any training (control practices).' They will also be told whether their practice did or did not receive the enhanced communication skills training. Efforts are made to mask researchers supporting patient data

**Table 1** Practitioner timelines

| | Allocation | | Postallocation (week) | | | | | On completing patient recruitment | On completing patient follow-up |
|---|---|---|---|---|---|---|---|---|---|
| Timepoint | 0 | +1 day | 1 | 2 | 3–8 | 8 | 34 | | |
| **Enrolment:** | | | | | | | | | |
| Eligibility screen | X | | | | | | | | |
| Informed consent | X | | | | | | | | |
| Site initiation visit | X | | | | | | | | |
| Allocation | | X | | | | | | | |
| **Interventions:** | | | | | | | | | |
| EMPathicO training | | | ▓ | ▓ | | | | | |
| No training (control) | | | ▓ | ▓ | | | | | |
| **Assessments:** | | | | | | | | | |
| Demographic and professional characteristics | X | | | | | | | | |
| Self-efficacy for empathy and optimism | X | | | | | X | X | | |
| Expectations, intentions for EMPathicO skills * | | | | X | | X | X | | |
| Practitioner-reported other training | | | | | | X | X | | |
| Qualitative interview | | | | | | | | X | X* |
| **Patient recruitment** | | | | | | | | | |
| Prepare invitations | | | ▓ | ▓ | | | | | |
| Recruit patients | | | | | ▓ | ▓ | | | |

*Intervention-arm practitioners only.

collection from intervention allocation; for example, the researchers collecting patient outcomes are not the same researchers who liaise with practices about the intervention. Efforts are made to mask practitioners so they are not aware of which specific patients are taking part; for example, the patient's PIS includes the instruction to 'please do not discuss your participation in the study with your GP, nurse, physiotherapist or any other primary care practitioner'. In the unlikely event that patient unblinding is deemed necessary for patient care, this will be done by the general practice and notified to the research team.

## Data analysis
### Data management
Web-based questionnaire data are stored securely on Qualtrics servers (see https://www.qualtrics.com/security-statement/). Questionnaire data collected by telephone or paper are entered into Qualtrics by one researcher and checked for accuracy by a second researcher.

Personal data are stored on a secure server at the University of Southampton in compliance with the General Data Protection Regulations and the Data Protection Act 2018.

## Statistical methods
Musculoskeletal and all-comers groups will be analysed separately. For the two primary outcomes, a linear mixed model will use all the observed data and implicitly assume that missing outcome scores are missing at random given the observed data. The BPI and PEI will be reported and analysed using postintervention scores, adjusting for baseline scores. The primary analyses for the BPI and PEI scores will be performed using a generalised linear mixed model framework with observations at 3 days, 1. 3 and 6 months (level 1) nested in participants (level 2) and participants nested in practices (level 3). Unadjusted results will be reported, as well as results adjusting for baseline values, stratification variables and other covariates as appropriate. As there may not be a constant treatment effect over time, a treatment–time interaction will be modelled and included if significant, with time treated as a random effect. An unstructured covariance matrix will be used. For secondary outcomes, the analyses will use a similar modelling approach, with mixed logistic and linear regression models as appropriate, a random effect for practice and controlling for baseline values, stratification variables and potential

**Table 2** Patient timelines

| | Enrol | Consultation | Postconsultation | | | |
|---|---|---|---|---|---|---|
| Timepoint | <−7 days | 0 | <7 days | +1 month | +3 months | +6 months |
| Enrolment: | | | | | | |
| Eligibility screen | X | | | | | |
| Informed consent | X | | | | | |
| Assessments: | | | | | | |
| Primary outcomes | | | | | | |
| Pain intensity | X | | X | X | X | X |
| Patient enablement | | | X | X | X | X |
| Secondary Outcomes | | | | | | |
| Global impression of symptom severity | X | | X | X | X | X |
| Global impression of symptom change | | | X | X | X | X |
| Pain interference | | | | X | | X |
| Patient satisfaction | | | X | | | |
| Health economics: EuroQol-five dimension-give level and ICEpop CAPability measure for Adults | X | | | X | | X |
| Adverse events | | | | X | X | X |
| Healthcare utilisation | X | | | | X | X |
| Prescribed medications, personal expenses and productivity | | | | | X | X |
| Process measures | | | | | | |
| Perceptions of empathy | | | X | | | |
| Perceptions of optimism | | | X | | | |
| Treatment expectations | | | X | | | |
| Anxiety | | | X | | | |
| Continuity of care | | | X | | | |
| Depression | | | X | | | |
| Sociodemographic characteristics | X | | | | | |
| Health characteristics | | | X | | | |
| Qualitative interview | | | X | | | X |

confounders. There are no formal preplanned subgroup analyses.

The intention to treat analysis (as randomised) will be undertaken regardless of any practice-level non-adherence to the intervention. All available data will be used, with a sensitivity analysis using multiple imputations if appropriate. Linear mixed models and multiple imputations both assume the data are missing at random; therefore, sensitivity analyses to data missing not at random will also be explored. A full and detailed statistical analysis plan will be developed prior to the final trial analysis and approved by TSC.

Interim analyses of outcomes are deemed unnecessary in this low-risk trial.

### Health economic analysis

An NHS perspective will be taken in the primary analysis; a wider perspective will be taken in the secondary analyses, including impacts on patients and productivity. The analysis will be intention to treat. Relevant covariates, including baseline EQ-5D-5L, potentially skewed data and the cluster design will be accounted for using appropriate regression models.[66] Cost-consequences will tabulate costs from each perspective for a range of outcomes. Cost-effectiveness will be estimated in a cost-utility analysis combining quality-adjusted life year and NHS costs. The incremental net monetary benefit statistic will be presented at standard NICE thresholds and if appropriate, incremental cost-effectiveness ratios will be estimated. Uncertainty will be addressed by bootstrapping, plotting cost-effectiveness acceptability curves and sensitivity analyses.

### Process analysis

Process analysis will focus on mechanisms of impact and test hypotheses derived from the logic model about relationships among variables, including mediators and moderators. Intervention usage data, captured by LifeGuide, will be incorporated using the AMUsED framework for Analysing and Measuring Usage and Engagement Data.[84]

### Qualitative and mixed-methods analysis

EMPathicO's potential impact post-trial will be evaluated using the RE-AIM framework to explore Reach, Effectiveness, Adoption, Implementation and

**Table 3** Qualitative and mixed-methods data analysis to evaluate intervention

| RE-AIM | Data source | Analysis |
|---|---|---|
| Reach | Management data | Proportion and characteristics of practitioners and patients taking part. Reasons for declining. |
| Effectiveness | All-comers group | Apply analysis plan from main trial to test intervention effectiveness in all-comers group. |
| | Qualitative data (patients and practitioners) | Compare experiences of EMPathicO across in-person, telephone and video consultations, and for musculoskeletal pain versus other conditions (framework analysis). |
| Adoption | Management data | Proportion and characteristics of invited practices taking part. Reasons for declining. |
| Implementation | LifeGuide usage and qualitative data | Assess patterns of usage and 'effective engagement' with EMPathicO. Explore barriers and facilitators to implementation in practice, drawing on Normalisation Process Theory[88] (framework analysis). |
| Maintenance | Qualitative data (patients and practitioners) | Explore opportunities to embed EMPathicO in existing training structures. Examine long-term maintenance of practitioner behaviour change and effects on patients (reflexive thematic analysis). |

Maintenance.[32 33] Drawing on data from the main trial, the all-comers group and the qualitative interviews, we will assess EMPathicO against the RE-AIM components using the approaches described in table 3.

### Ethics and dissemination

#### Safety, adverse events and insurance

This trial is classified as low-risk following a risk assessment, and there are no provisions for post-trial care. The team does not expect any adverse events (untoward medical occurrence in a trial participant) or serious adverse events (that result in death, are life-threatening, require hospitalisation or prolongation of existing hospitalisation, result in persistent or significant disability or incapacity or consist of a congenital anomaly, birth defect or other medically important condition). However, adverse events are being collected (primarily via self-report), recorded and reported where necessary in accordance with the principles of ICH Good Clinical Practice and the requirements of the research ethics committee, sponsor and TSC.

Individual practitioners are responsible for maintaining appropriate cover with a medical defence organisation. University of Southampton insurance may also apply where the cause of harm was not due to clinical negligence.

#### Approvals, oversight and monitoring

The sponsor is the University of Southampton (rgoinfo@soton.ac.uk). Approval was received from the South Central Hampshire B Research Ethics Committee on 1 July 2022 and from the Health Research Authority and Health and Care Research Wales on 6 July 2022 (REC reference 22/SC/0145; IRAS project ID 312208). Protocol amendments are submitted for approval as required to the study sponsor and ethics committee and notified, where necessary, to all those concerned.

The TSC provides trial oversight and advice through its independent chairperson to the Trial Management Group and the funder on all aspects of the trial. The TSC assumes the responsibilities of the Data Monitoring Committee and reviews information on progress and accruing data. Online supplemental material 4 presents the TSC Charter. Online supplemental material 5 presents stopping criteria. Annual and interim progress reports are submitted to the funder.

#### Dissemination

Patient recruitment commenced on 16 November 2022 and is ongoing at the time of manuscript submission. Results will be communicated to participants and disseminated to academic, practitioner and public audiences via peer-reviewed journal articles, conferences and other appropriate formats, for example, blogs. Our public collaborators will co-lead dissemination activities. Results will be reported in accordance with Consolidated Standards of Reporting Trials guidelines extensions for cluster-randomised trials[85] and trials of non-pharmacological interventions[86] and the American Psychological Association Journal Article Reporting Standards for qualitative and mixed-methods research.[87] We will adhere to the International Committee of Medical Journal Editors (https://www.icmje.org/) criteria for authorship and use the CRediT taxonomy (https://credit.niso.org/). Online supplemental material 6 summarises data access plans.

**Author affiliations**
[1]School of Psychology, University of Southampton, Southampton, UK
[2]Primary Care Research Centre, School of Primary Care, Population Science, and Medical Education, University of Southampton, Southampton, UK
[3]Centre of Academic Primary Care, Bristol Medical School, University of Bristol, Bristol, UK
[4]Keele School of Medicine, Keele University, Newcastle-under-Lyme, UK
[5]Population Health Sciences, University of Bristol Faculty of Health Sciences, Bristol, UK

6Wolfson Institute of Population Health, Queen Mary University of London, London, UK
7Health Economics Bristol, Population Health Sciences, University of Bristol, Bristol, UK
8Unit of Academic Primary Care, University of Warwick, Coventry, UK
9Leicester Medical School, University of Leicester, Leicester, UK
10Faculty of Philosophy, University of Oxford, Oxford, UK
11Southampton Clinical Trials Unit, University of Southampton and University Hospital Southampton NHS Foundation Trust, Southampton, UK

**Acknowledgements** NIHR Local Clinical Research Networks (CRNs) supported practice recruitment.

**Contributors** Allocated using CRediT categories. Conceptualisation (study idea) and funding acquisition: HE, FLB, JH, PL, BS, GML, LM, JV, JB, CM, LC, MJR, KG and HA. Methodology (designing, planning and developing study methods): FLB, HE, PL, GML, BS, LM, JV, CM, LC, MJR, KG, JH, HA, JB, NC, ET, SP, RDH, JN, NI, PHL, TB, AH and MER. Investigation (data collection): NC, RDH, ET, AH, MER and SP. Data curation (study management data and data cleaning): NC, RDH, ET, AH, MER and SP. Project administration (managing and co-ordinating research activity plans and execution): FLB, HE and NC. Software (implementation and support for the e-learning intervention): SP. Supervision (oversight, leadership and mentorship): FLB and HE. Visualisation (creation and presentation of figures): SP, LM and FLB. Writing (original draft): HE, FLB, JH, BS, TB, MJR, KG, HA and JB. Writing (review, revisions and editing): FLB, HE, PL, GML, BS, LM, JV, CM, LC, MJR, KG, JH, HA, JB, NC, ET, SP, RDH, JN, NI, PHL, TB, AH and MER.

**Funding** This project is funded by the National Institute for Health Research (NIHR) School for Primary Care Research grant (project reference 563). The Primary Care Research Centre, University of Southampton is a member of the NIHR School for Primary Care Research and supported by NIHR Research funds. Service support costs will be paid by the CRN. CM is funded by the National Institute for Health Research (NIHR) Collaborations for Leadership in Applied Health Research and Care West Midlands (grant number N/A) and the NIHR School for Primary Care Research. The EMPathicO e-learning tool was developed using LifeGuide software, which was partly funded by the National Institute for Health Research Southampton Biomedical Research Centre (BRC) (grant number N/A). The views expressed are those of the authors and not necessarily those of the NIHR or the Department of Health and Social Care. The study sponsor (University of Southampton) and funders (NIHR SPCR) have no role in study design; collection, management, analysis, and interpretation of data; writing of the report or the decision to submit the report for publication.

**Competing interests** All authors have completed the ICMJE uniform disclosure form at http://www.icmje.org/disclosure-of-interest/ and declare: financial support for the submitted work from the NIHR; CDM is Director of the NIHR School for Primary Care Research; HA has received research grants from NIHR and Research Council of Norway, payment for delivering lecture to GPs in training about remote consultations, travel expenses to attend Scientific Foundation Board meeting; HA is chair of a steering committee at University of Leeds, member of advisory boards at Imperial College London and University of Manchester, and vice-chair of the Scientific Foundation Board Royal College of General Practitioners; HA is Officer at Prof Andrew Beggs Ltd. No other relationships or activities that could appear to have influenced the submitted work.

**Patient and public involvement** Patients and/or the public were involved in the design, or conduct, or reporting, or dissemination plans of this research. Refer to the Methods section for further details.

**Patient consent for publication** Not applicable.

**Ethics approval** South Central – Hampshire B Research Ethics Committee and Heath Research Authority and Health and Care Research Wales (REC reference 22/SC/0145; IRAS project ID 312208).

**Provenance and peer review** Not commissioned; externally peer reviewed.

**ORCID iDs**
Felicity L Bishop http://orcid.org/0000-0002-8737-6662
Nadia Cross http://orcid.org/0000-0002-4148-7180
Rachel Dewar-Haggart http://orcid.org/0000-0002-3757-1152
Emma Teasdale http://orcid.org/0000-0001-9147-193X
Amy Herbert http://orcid.org/0009-0008-6109-6006
Michelle E Robinson http://orcid.org/0000-0002-2266-8250
Matthew J Ridd http://orcid.org/0000-0002-7954-8823
Christian Mallen http://orcid.org/0000-0002-2677-1028
Lorna Clarson http://orcid.org/0000-0003-0828-9649
Jennifer Bostock http://orcid.org/0000-0001-9261-9350
Taeko Becque http://orcid.org/0000-0002-0362-3794
Beth Stuart http://orcid.org/0000-0001-5432-7437
Kirsty Garfield http://orcid.org/0000-0002-8301-3602
Leanne Morrison http://orcid.org/0000-0002-9961-551X
Sebastien Pollet http://orcid.org/0000-0001-9924-9225
Jane Vennik http://orcid.org/0000-0003-4602-9805
Helen Atherton http://orcid.org/0000-0002-7072-1925
Jeremy Howick http://orcid.org/0000-0003-0280-7206
Geraldine M Leydon http://orcid.org/0000-0001-5986-3300
Jacqui Nuttall http://orcid.org/0000-0002-5826-2594
Nazrul Islam http://orcid.org/0000-0003-3982-4325
Paul H Lee http://orcid.org/0000-0002-5729-6450
Paul Little http://orcid.org/0000-0003-3664-1873
Hazel A Everitt http://orcid.org/0000-0001-7362-8403

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
