## [Reviewer comments · BMJ Open]

ARTICLE DETAILS

TITLE (PROVISIONAL)	Talking in Primary Care (TIP): Protocol for a cluster-randomised controlled trial in UK primary care to assess clinical and cost effectiveness of communication skills e-learning for practitioners on patients' musculoskeletal pain and enablement
AUTHORS	Bishop, Felicity; Cross, Nadia; Dewar-Haggart, Rachel; Teasdale, Emma; Herbert, Amy; Robinson, Michelle; Ridd, Matthew; Mallen, Christian; Cl arson, Lorna; Bostock, Jennifer; Becque, Taeko; Stuart, Beth; Garfield, Kirsty; Morrison, Leanne; Pollet, Sebastien; Vennik, Jane; Atherton, Helen; Howick, Jeremy; Leydon, Geraldine; Nuttall, Jacqui; Islam, Nazrul; Lee, Paul; Little, Paul; Everitt, Hazel

VERSION 1 – REVIEW

REVIEWER	B Heleno Universidade Nova de Lisboa, Family Medicine Unit
REVIEW RETURNED	21-Dec-2023

GENERAL COMMENTS	Thank you for the opportunity to review this protocol of an ongoing study, which began recruiting participants in November 2022. The introduction of the protocol is effectively concise, offering a clear background on the role of communication in patient outcomes. It well delineates what is currently known and unknown in this field, aptly justifying the study's aims and objectives. The methodological approach is robust, utilizing a cluster-randomized trial to assess both effectiveness and cost-effectiveness. The protocol includes qualitative data collection from both patients and clinicians. It also uses a recognized framework for implementation analysis in this complex intervention. The description of the cluster-randomized trial is thorough and suggests a potential for low risk of bias. The processes related to randomization, participant identification and recruitment, adherence to intended interventions, and outcome measurement appear to be well-considered and low-risk in terms of bias. Strategies to minimize participant and cluster loss are in place. I recommend enhancing clarity in two specific areas: a) The sequence of practice, practitioner, and participant recruitment in relation to cluster randomization. My understanding is that practice recruitment precedes the randomization of clusters, while patient recruitment follows; but this could be explicitly stated. I only understood that practitioners are recruited before cluster randomization, after reading the informed consent.
--

	b) While the outcome domains (pain intensity and patient enablement) are clearly defined, it was not entirely clear to me whether the report will include post-intervention scores or change scores for the BPI and the PEI. The description of the Empathico intervention is thorough, as detailed in reference 24, aligning well with the 'Template for intervention description and replication' checklist. It would be valuable, however, to know whether additional strategies such as persuasion, incentivization, or modeling are also part of this complex intervention. Furthermore, the protocol describes two trials in two different populations (those with musculoskeletal pain, and the "all-comer" group). It would be beneficial for readers to know whether the researchers anticipate the intervention to be effective in both populations. Such clarification would aid in interpreting the final results, especially in the event of discordant findings between the musculoskeletal pain and the "all-comer" groups. Overall, this is a well-structured and promising study protocol. I look forward to seeing the contributions of its findings to the field.
--	--

REVIEWER	Katherine A. Pohlman Parker University, Research Center
REVIEW RETURNED	02-Jan-2024

GENERAL COMMENTS	Thanks for conducting this important research. Looking forward to seeing the results of the study. Please find below some minor suggestions to help with the clarity of the study. PPI - please add more details regarding the diversity of backgrounds and who the bimonthly meetings are run (i.e. virtually for 1-hour) Target population - Add details regarding the NHS definition/acceptance of general practices, including the lack of inclusion of osteopathic physicians and chiropractors. Patients with Musculoskeletal Pain Eligibility Criteria - add more details on who the 'musculoskeletal pain group' are. I found the 'e.g.' for the musculoskeletal pain confusing with only 'knee' stated as an extremity - either add more extremities or just state 'upper and lower' extremity. State how the use or just that the use of proxies will be documented. Text with Figure 1 - unclear if this is for the provider or the patient. Figure 2 starts out focused on osteoarthritis - prior to this point was focused on musculoskeletal pain. Please review for consistency or clear explanation for the change. Patient Recruitment - Please describe if this will be consecutive patients or something else. How will patients who decline participation be recorded? Questionnaires, Data Collection and Participant Retention - Second paragraph. Instead of 'incentive', maybe explain as a reimbursement for time spent completing the questionnaires. Blinding - how will the patients be blinded? Will there be any de-
---

	briefing. Please add details about what the providers in the usual care group know about the study design - was a Zelen design considered? Add clarity that patients are asked to not share their study participation with their provider. Safety, Adverse Events, and Insurance - add details regarding what the 'accordance with good clinical practice' for those outside of England.
--	--

VERSION 1 – AUTHOR RESPONSE

Reviewer: 1	
The sequence of practice, practitioner, and participant recruitment in relation to cluster randomization. My understanding is that practice recruitment precedes the randomization of clusters, while patient recruitment follows; but this could be explicitly stated. I only understood that practitioners are recruited before cluster randomization, after reading the informed consent.	Clarifying text added to Methods / Design Section: General practices constitute the clusters; practices are recruited and then randomised 1:1 EMPathicO: control. [...] Patient recruitment commences at least two weeks after the GP site is randomised (enabling time for intervention sites to complete the intervention training whilst maintaining consistent set up timelines across both arms). All adults (18+); verbally consulting a participating practitioner are invited to participate in the trial (see exclusions below).
While the outcome domains (pain intensity and patient enablement) are clearly defined, it was not entirely clear to me whether the report will include post-intervention scores or change scores for the BPI and the PEI.	Change scores are not recommended as they are likely to comprise regression to the mean, measurement error and contextual effects, in addition to the intervention effect. We have added this detail to: “The BPI and PEI will be reported and analysed using post-intervention scores, adjusting for baseline score.”
The description of the Empathico intervention is thorough, as detailed in reference 24, aligning well with the 'Template for intervention description and replication' checklist. It would be valuable, however, to know whether additional strategies such as persuasion, incentivization, or modeling are also part of this complex intervention.	The BCW intervention functions and specific BCTs included in Empathico are described in supplementary material to reference 24, available online open access. A brief summary has been added to the manuscript: “Using the Behaviour Change Wheel, EMPathicO was designed to target users’ motivation (reflective, autonomic), capability (physical, psychological), and opportunity (environmental), through intervention functions of persuasion, incentivization, enablement, education, training, modelling, and environment restructuring. Multiple Behaviour Change Techniques were used to achieve these functions, including demonstration, information provision, goal-setting, action planning, and instruction. For a complete behavioural analysis of EMPathicO see supplementary material in our intervention development paper. ²⁴ “
Furthermore, the protocol describes two trials in two different populations (those with musculoskeletal pain, and the "all-comer" group). It would be beneficial for readers to	Explanatory text added to aims and objectives: This clinical focus on musculoskeletal pain was

know whether the researchers anticipate the intervention to be effective in both populations. Such clarification would aid in interpreting the final results, especially in the event of discordant findings between the musculoskeletal pain and the "all-comer" groups.	chosen to align with the EMPathicO training, which includes modules on clinical empathy, realistic optimism, and how to communicate these better in the context of consultations for osteoarthritis. Including a condition-specific module permitted clear demonstration of communication skills in a particular context, which made the training better targeted and potentially more effective.³¹ A painful musculoskeletal condition was chosen because much (but not all) of the evidence that underpins the importance of clinical empathy and realistic optimism for patient outcomes is derived from studies of pain and painful conditions; osteoarthritis was chosen because it is a prevalent painful musculoskeletal condition in primary care. Secondary objectives are:  • To estimate EMPathicO’s cost-effectiveness and effects on patient-reported quality of life and other secondary outcomes, over 6 months from index consultation, in patients with musculoskeletal pain. • To test hypothesised mechanisms of action. • To explore EMPathicO’s potential for implementation, by:  o Determining EMPathicO’s effects on patient enablement, patient-reported quality of life and other secondary outcomes over 6 months from index consultation, in patients ineligible for the musculoskeletal pain group (i.e., presenting with other symptoms and/or very low levels of musculoskeletal pain, hereafter referred to as ‘all-comers’). This group was included because clinical empathy and realistic optimism may be beneficial for many different symptoms seen in primary care, and when practitioners adopt new communication behaviours within consultations for one type of condition these skills may ‘spill-over’ and also be implemented in consultations for other conditions. We wanted to evaluate any such additional benefits.
Reviewer: 2	
PPI - please add more details regarding the diversity of backgrounds and how the bimonthly meetings are run (i.e. virtually for 1-hour)	We have expanded our description of PPIE: “We continue working closely with our Patient Advisory Group, led by our PPIE lead, JB, who sits on our trial management group. This group comprises six patient and public contributors of varying ages, ethnic

	backgrounds (three from Black and Minority Ethnic backgrounds, three from White backgrounds), gender (three female, three male), and geographical locations within England. One member is neurodivergent and all have lived experience of MSK pain as patients or carers. Our panel meet virtually for one hour bimonthly...”
Target population - Add details regarding the NHS definition/acceptance of general practices, including the lack of inclusion of osteopathic physicians and chiropractors.	Definition of general practices added and attributed to the General Practice Workforce Data publication which provides a searchable dashboard for readers to explore the composition of the general practice workforce. “Eligible: NHS general practices in England and Wales, where a general practice is “an organisation which offers Primary Care medical services by a qualified General Practitioner who can prescribe medicine and where patients can be registered and held on a list.”³⁵” Criteria for practitioners clarified: “Eligible: practitioners from any discipline who are working within participating GP surgeries and seeing patients with musculoskeletal pain (e.g., GPs, Practice Nurses, Physiotherapists, Pharmacists, Physician Associates).” Osteopaths and chiropractors, like other healthcare providers (e.g., acupuncturists), are eligible for the study if they meet the inclusion criteria.
Patients with Musculoskeletal Pain Eligibility Criteria - add more details on who the 'musculoskeletal pain group' are. I found the 'e.g.' for the musculoskeletal pain confusing with only 'knee' stated as an extremity - either add more extremities or just state 'upper and lower' extremity.	Amended as suggested: “musculoskeletal pain (e.g. back, hip, upper/lower extremity, neck pain - consistent with ICD-11’s diseases of the musculoskeletal system ³⁵);”
State how the use or just that the use of proxies will be documented.	Added: “People without English as a first language are eligible, interpreters are available to support access to trial paperwork and patient-reported measures, and their use is recorded”
Text with Figure 1 - unclear if this is for the provider or the patient.	Added: “The brief interactive e-learning modules are completed by practitioners and can be completed separately or together in less than 75 minutes and cover clinical empathy, realistic optimism, tailoring

	empathy and optimism for patients with osteoarthritis (a common cause of musculoskeletal pain), evaluating one's own consultations, and goal-setting. Figure 1 summarises the structure and contents of the modules."
Figure 2 starts out focused on osteoarthritis - prior to this point was focused on musculoskeletal pain. Please review for consistency or clear explanation for the change.	Additional text in Aims/Objectives explains this: "This clinical focus on musculoskeletal pain was chosen to align with the EMPathicO training, which includes modules on clinical empathy, realistic optimism, and how to communicate these better in the context of consultations for osteoarthritis. Including a condition-specific module permitted clear demonstration of communication skills in a particular context, which made the training better targeted and potentially more effective.³¹ A painful musculoskeletal condition was chosen because much (but not all) of the evidence that underpins the importance of clinical empathy and realistic optimism for patient outcomes is derived from studies of pain and painful conditions; osteoarthritis was chosen because it is a prevalent painful musculoskeletal condition in primary care." Figure 2 has also been modified to clarify and improve alignment.
Patient Recruitment - Please describe if this will be consecutive patients or something else. How will patients who decline participation be recorded?	Practices invite consecutive patients consulting participating practitioners within the recruitment period, after screening out any patients who do not have capacity for consent, or where there are medical grounds for excluding the patient (e.g., very unwell generally, severe mental distress). And The number of patient invitation emails/texts sent by each site is collected and recorded centrally. Qualtrics records instances of patients accessing the study website but declining consent and/or not meeting inclusion criteria.
Questionnaires, Data Collection and Participant Retention - Second paragraph. Instead of 'incentive', maybe explain as a reimbursement for time spent completing the questionnaires.	Labelling these vouchers as reimbursement or otherwise payment for time spent could have tax and other related implications. Therefore, we retain the 'incentive' nomenclature.
Blinding - how will the patients be blinded? Will there be any de-briefing. Please add details about what the providers in the usual care group know about the study design -	In this cluster-randomised trial randomisation happens at the level of general practice, not at the level of individual practitioner or individual patient. Additional

was a Zelen design considered? Add clarity that patients are asked to not share their study participation with their provider.	detail has been added to the Blinding section: Patients and the trial statistician are masked to intervention allocation. Patients are not told in the PIS that as part of this study their general practice has been randomly allocated to intervention or control. This was approved by the ethics committee and is appropriate in this cluster-randomised trial where the communication-skills training intervention is very low risk and within the broad scope of usual practice. After all data collection is complete, patients will be debriefed in writing (email/mail) and told that “at the start of the TIP study some of the GP practices taking part had communication skills training (intervention practices) and some GP practices did not have any training (control practices).” They will also be told whether their practice did or did not receive the enhanced communication skills training. Efforts are made to mask researchers supporting patient data collection to intervention allocation; for example, the researchers collecting patient outcomes are not the same researchers who liaise with practices about the intervention. Efforts are made to mask practitioners to which patients are taking part; for example, the patient PIS includes the instruction to “please do not discuss your participation in the study with your GP, nurse, physiotherapist, or any other primary care practitioner”.
Safety, Adverse Events, and Insurance - add details regarding what the 'accordance with good clinical practice' for those outside of England.	However, adverse events are being collected (primarily via self-report), recorded and reported where necessary in accordance with the principles of ICH Good Clinical Practice and the requirements of the research ethics committee, sponsor, and trial steering committee.